# SyNC: Balancing Fidelity and Diversity of Synthetic Data Representations in CLIP-based Few-Shot Learning via Neural Collapse

## Abstract

In few-shot learning, augmenting real data with synthesized images from text-to-image diffusion models has emerged as a promising direction. Although numerous studies have been proposed to improve the performance of this training framework, they often fail to adequately address the critical trade-off between fidelity and diversity when training with synthetic data. In this work, we propose SyNC, a novel training paradigm that explicitly balances these characteristics in the feature space through two complementary mechanisms. First, we leverage an optimal geometric prototype structure built upon the Neural Collapse phenomenon to increase fidelity, guiding the representations of both real and synthetic data toward their corresponding equiangular tight frame (ETF) prototypes. Second, we introduce an innovative regional contrastive loss function specifically designed to enhance diversity by improving the distinction between misclassified synthetic data features, thereby encouraging more varied and robust representations. Extensive experimental results demonstrate the effectiveness of our proposed method, which outperforms state-of-the-art approaches on average across few-shot image classification benchmarks and shows significant improvements on fine-grained datasets. Further analysis demonstrates that our method achieves a more favorable balance between representation fidelity and diversity, revealing a correlation between these factors and overall model performance.

## 1 Introduction

Deep learning has achieved remarkable performance when sufficient annotated data is available (He et al., 2016; van den Oord et al., 2016; Wu et al., 2016). However, real-world scenarios often present limited labeled training data, making Few-Shot Learning (FSL) a critical research area for developing models that can learn effectively from minimal samples. Recent advances in generative modeling have established synthetic data as a valuable resource for training deep learning models in both computer vision (Yuan et al., 2024; Li et al., 2025) and natural language processing (Luo et al., 2025; Gan & Liu, 2025). This development has naturally led to incorporating synthetic data into few-shot learning frameworks to address the fundamental challenge of data scarcity.

In the vision-language domain, few-shot learning with CLIP-based models (Radford et al., 2021) has garnered significant attention due to their remarkable generalization capabilities. Consequently, researchers have explored synthetic data augmentation for few-shot CLIP learning from multiple perspectives: modifying real samples as generator inputs (He et al., 2023; da Costa et al., 2023), integrating self-supervised learning knowledge (Zhang et al., 2023; Haoyuan et al., 2025), and employing distributional matching with theoretical guarantees (Kim et al., 2024; Nguyen et al., 2025b).

A fundamental challenge when training with synthetic data lies in balancing the dataset quality and diversity. This trade-off has been extensively studied in data curation for large language models (Liu et al., 2024; Qin et al., 2025; Wu et al., 2025), multimodal learning (Goyal et al., 2024; Wang et al., 2024), and synthetic data for instruction tuning (Li et al., 2023; Yu et al., 2023; Nguyen et al., 2025a). This motivates us to ask a critical question for few-shot learning with synthetic data: *How can we balance the quality of synthesized images for accurate training while maintaining sufficient diversity to achieve effective model generalization?*

Existing CLIP-based few-shot learning methods with synthetic data have only partially addressed this trade-off, often focusing on one aspect at the expense of the other. DataDream (Kim et al., 2024) and ProtoAug (Nguyen et al., 2025b) tackle the quality problem by matching real and synthetic distributions at pixel and feature representation levels, but address diversity only through reduced diffusion model guidance scales, a limited approach that may compromise generation quality. Conversely, DISEF (da Costa et al., 2023) and ImagineFSL (Haoyuan et al., 2025) enhance diversity through hard prompt-tuning techniques and detailed prompt generation via external image captioning or large language models. However, this increased diversity often pushes synthetic images further from the real data distribution. While these methods attempt self-correction through CLIP filtering, such filtering remains inaccurate and suffers from false positives (samples with poor semantic alignment that nonetheless achieve high CLIP scores) (Mahmoud et al., 2024). Additionally, methods like CaFo (Zhang et al., 2023) and ImagineFSL (Haoyuan et al., 2025) incorporate self-supervised learning paradigms, introducing significant computational overhead to an already compute-intensive image synthesis process.

To address the aforementioned challenges, we propose SyNC, an innovative and universal training paradigm to balance the fidelity and diversity of **Sy**nthetic data representations in CLIP-based Few-Shot Learning via **N**eural **C**ollapse.

Our first contribution leverages the Neural Collapse (NC) phenomenon (Papyan et al., 2020), which describes the optimal geometric structure that emerges in deep network representations during training. We design a loss function that encourages the representations of both real and synthetic samples to converge toward the same optimal prototype of an equiangular tight frame (ETF) structure. This approach ensures training quality by not only aligning real and synthetic representations with each other, but more importantly, guiding them toward the theoretically optimal classification geometry structure. Unlike previous methods that rely only on distribution matching, our proposed method provides a principled approach to achieving high-fidelity representations.

Our second contribution addresses the diversity challenge through a novel regional contrastive loss that enhances inter-class separability while maintaining intra-class cohesion. This loss component employs an elegant modification of supervised contrastive loss that specifically targets misclassified synthetic samples, pushing representations of different classes further apart in feature space. Our regional contrastive formulation integrates seamlessly into the CLIP training paradigm.

Our contributions can be summarized as follows:

- We propose SyNC, a novel training paradigm that explicitly balances fidelity and diversity of feature representation when training with both real and synthetic data, addressing the fundamental trade-off that previous methods only partially resolve.

- We design two complementary loss functions to address the fidelity-diversity trade-off: a Neural Collapse loss that leverages optimal geometric structures to improve representation quality and alignment, and a regional supervised contrastive loss that enhances diversity by targeting misclassified synthetic samples.

- We provide extensive experimental validation showing that our method outperforms state-of-the-art approaches across few-shot fine-tuning image classification benchmarks, with substantial gains on fine-grained datasets. Comprehensive ablation studies reveal the correlation between quality-diversity balance and overall model performance, demonstrating the effectiveness of our method and underlying the proposed mechanisms.

## 2 RELATED WORK

### 2.1 SYNTHETIC DATA AS AUGMENTATION

Recent advances in generative models have established synthetic data as a valuable augmentation strategy across multiple domains. In computer vision, early approaches focused on aligning synthetic and real data distributions through text-prompt engineering (He et al., 2023; Lei et al., 2023; Sariyildiz et al., 2022). To achieve better alignment, RealFake (Yuan et al., 2024) minimizes maximum mean discrepancy between real and synthetic distributions through generator fine-tuning. Gen-DataAgent (Li et al., 2025) advances this paradigm by enhancing diversity via caption perturbation

and improving quality through Variance of Gradient (VoG) score filtering during training. In natural language processing, synthetic data augmentation has been widely adopted for large language models, initially targeting zero-shot and few-shot settings (Meng et al., 2022; Li et al., 2023). Subsequent work has emphasized controllable generation to mitigate hallucination and ensure high quality through multi-step data filtering and self-correction meta-prompts. These approaches further enhance diversity by leveraging combinations of different attributes and arithmetic concepts (Gupta et al., 2024; Dekoninck et al., 2024; Huang et al., 2025).

## 2.2 Few-shot Learning with CLIP-based Models

Contrastive Language Image Pretraining (CLIP) (Radford et al., 2021) has emerged as a promising solution for few-shot image classification due to its strong generalization ability. A traditional and efficient way to adapt CLIP to downstream tasks is prompt tuning (Jia et al., 2022; Khattak et al., 2023; Zheng et al., 2024; Hao et al., 2025; Liu et al., 2025) or adapter tuning (Cheng et al., 2023; Yang et al., 2024). With the increasing performance of generative models, a promising direction is augmenting the amount of few-shot data with synthesized data. IsSynth (He et al., 2023) and DISEF (da Costa et al., 2023) add noise to few-shot samples before processing through generative models, but this creates distribution gaps and limits diversity. CaFo (Zhang et al., 2023) combines pretraining knowledge from four generative models, while DataDream (Kim et al., 2024) fine-tunes generative models for better distribution matching. ProtoAug (Nguyen et al., 2025b) additively matches synthetic and real distributions, and ImagineFSL (Haoyuan et al., 2025) transfers knowledge from purely synthetic samples, further demonstrating the potential of generative data.

## 2.3 Neural Collapse

Papyan et al. (2020) reveals the neural collapse phenomenon, where last-layer features converge to their within-class means, and these means along with classifier vectors collapse to the vertices of a simplex equiangular tight frame during the terminal phase of training on balanced datasets.

**Definition 2.1** (Simplex Equiangular Tight Frame). A collection of vectors $\mathbf{m}_i \in \mathbb{R}^D$, $i = 1, 2, \cdots, N; D \geq N - 1$, is said to be a simplex equiangular tight frame if:

$$\mathbf{M} = \sqrt{\frac{N}{N-1}} \mathbf{U} \left( \mathbf{I}_N - \frac{1}{N} \mathbf{1}_N \mathbf{1}_N^T \right), \tag{1}$$

where $\mathbf{M} = [\mathbf{m}_1, \cdots, \mathbf{m}_N] \in \mathbb{R}^{D \times N}$, $\mathbf{U} \in \mathbb{R}^{D \times N}$ allows a rotation and satisfies $\mathbf{U}^T \mathbf{U} = \mathbf{I}_N$, $\mathbf{I}_N$ is the identity matrix, and $\mathbf{1}_N$ is an all-ones vector.

All vectors in a simplex ETF have an equal $\ell_2$ norm and the same pair-wise angle, *i.e.*,

$$\mathbf{m}_i^T \mathbf{m}_j = \frac{N}{N-1} \delta_{i,j} - \frac{1}{N-1}, \forall i, j \in [1, N], \tag{2}$$

where $\delta_{i,j}$ equals 1 when $i = j$ and 0 otherwise. The pair-wise angle $-\frac{1}{N-1}$ is the maximal equiangular separation of $N$ vectors in $\mathbb{R}^D$ (Strohmer & Heath, 2003).

Then the neural collapse (NC) phenomenon can be formally described as:

**(NC1)** Within-class variability of the last-layer features collapse: $\Sigma_W \rightarrow \mathbf{0}$, and $\Sigma_W := \mathrm{Avg}_{i,n}(\mathbf{h}_{n,i} - \mathbf{h}_n)(\mathbf{h}_{n,i} - \mathbf{h}_n)^T$, where $\mathbf{h}_{n,i}$ is the last-layer feature of the $i$-th sample in the $n$-th class, and $\mathbf{h}_n = \mathrm{Avg}_i \mathbf{h}_{n,i}$ is the within-class mean of the last-layer features in the $n$-th class;

**(NC2)** Convergence to a simplex ETF: $\tilde{\mathbf{h}}_n = (\mathbf{h}_n - \mathbf{h}_G)/||\mathbf{h}_n - \mathbf{h}_G||, n \in [1, N]$, satisfies Eq. (2), where $\mathbf{h}_G$ is the global mean of the last-layer features, *i.e.*, $\mathbf{h}_G = \mathrm{Avg}_{i,n}\{\mathbf{h}_{n,i}\}$;

**(NC3)** Self duality: $\tilde{\mathbf{h}}_n = \mathbf{w}_n/||\mathbf{w}_n||$, where $\mathbf{w}_n$ is the classifier vector of the $n$-th class;

**(NC4)** Simplification to the nearest class center prediction: $\arg\max_n \langle \mathbf{h}, \mathbf{w}_n \rangle = \arg\min_n ||\mathbf{h} - \mathbf{h}_n||$, where $\mathbf{h}$ is the last-layer feature of a sample to predict for classification.

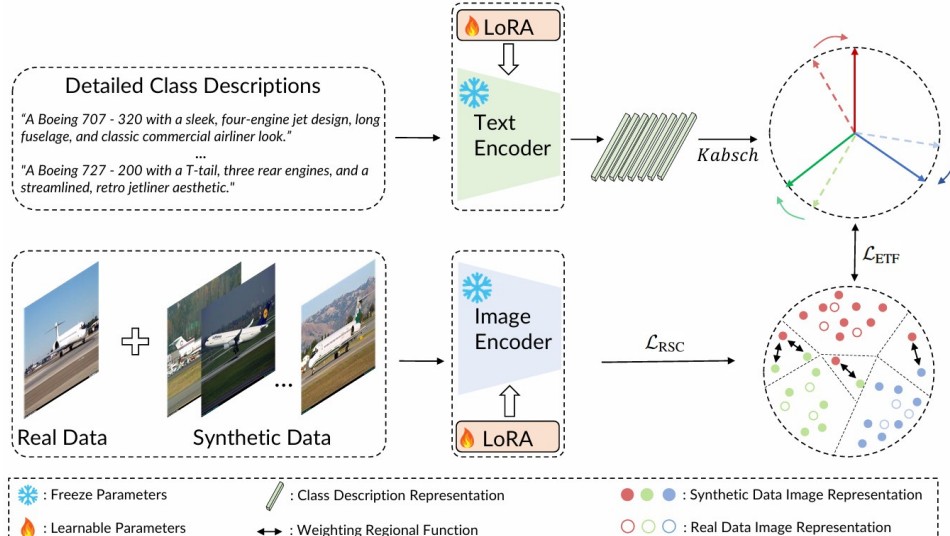

Figure 1: An overview of our SyNC framework: First, enhanced prompts are forwarded through the CLIP text encoder to obtain the predicted data distribution. Next, the ETF structure is aligned with resulting embeddings via the Kabsch algorithm to attain the optimal geometric class prototypes, as discussed in Section 3.1. Finally, both the text and image encoders are refined through LoRA fine-tuning, guided by the $\mathcal{L}_{\text{ETF}}$ (Eq. 4) and $\mathcal{L}_{\text{RSC}}$ (Eq. 5) loss components, as detailed in Sections 3.2 and 3.3.

# 3 METHODOLOGY

In this section, we demonstrate the construction of two loss components that improve the fidelity and diversity of feature representation: an ETF-based contrastive loss that aligns the representations with neural collapse prototypes and a regional supervised contrastive loss that improves diversity while maintaining class discrimination, before describing the overall training framework SyNC. The overall framework pipeline is illustrated in Figure 1.

## 3.1 NEURAL COLLAPSE AS PROTOTYPE LEARNING

Under the neural collapse solution, the final-layer features converge to the vertices of a simplex equiangular tight frame (ETF), acting as their class prototypes. An ETF belongs to the class of Grassmannian frames, known to attain minimal coherence relative to all unit-norm frames. This implies that the prototypes of NC structure achieve maximally pairwise distance. Consequently, numerous works have assigned their model's fixed class prototypes based on NC (Yang et al., 2022; 2023; Pham et al., 2025), achieving notable performance in continual learning for classification tasks. However, none of the mentioned methods have addressed the potential impact of Neural Collapse on the standard few-shot settings.

One of the main reasons why directly applying NC to standard few-shot settings can lead to performance degradation is the random initialization of the ETF structure. Since class classifiers are randomly initialized, their positions in Euclidean space may be substantially distant from the data distribution, hindering the alignment between the last-layer features and their corresponding class prototypes. As random initialization of the ETF structure can undermine model performance, it is natural to ask: why not initialize ETF in a way that is better aligned with the input feature distribution? To this end, we propose an adaptive mechanism that dynamically adjusts both the norm and the direction of the ETF prototype to align with the feature embedding space of its corresponding class via Kabsch algorithm (Kabsch, 1978).

Given a few-shot dataset $\mathcal{D}^{\text{fs}} = \{(x_i, y_i)\}_{i \in [m]}$, a synthesized dataset $\mathcal{D}^{\text{synth}} = \{(\hat{x}_j, \hat{y}_j)\}_{j \in [s]}$ and let $\mathcal{D} = \mathcal{D}^{\text{fs}} \cup \mathcal{D}^{\text{synth}}$, where $y_i \in \{1, 2, \ldots, N\}$, $m = N \times K$ denotes the number of real samples

and $s = N \times T$ represents the number of synthesized samples, with $N$ is the number of classes, $K$ is the number of few-shot samples per class, and $T$ is the number of synthesized samples per class.

To get the predicted distribution of the data, we leverage the representation of the class description of each label. Specifically, for each label $y_i \in \{1, 2, \ldots, N\}$, instead of using the naive prompt 'a photo of a [CLS]', we utilize enriched descriptions $Des_{y_i}$ generated by GPT-4 (OpenAI, 2023). The enhancing prompt process is detailed in Appendix C.2. These descriptions are then encoded by the pretrained CLIP text encoder to yield class-specific feature vectors. The collection of these vectors forms a matrix $T \in \mathbb{R}^{D \times N}$, where $D$ denotes the dimension of the CLIP embedding space. Then, the initial class prototype is constructed as a matrix $W_{\mathrm{ETF}} \in \mathbb{R}^{D \times N}$ according to the ETF structure described in Definition 2.1. In addition, the $l_2$ norm of each ETF prototype is equal to the average of the norm of the matrix $T$. Since its orientation and position are arbitrary prior to alignment, so in order to match the ETF classifiers with the semantic representation, we align $W$ to the data distribution $T$ by finding the optimal rotation matrix $R$. We employ the Kabsch algorithm to solve for the rigid transformation that minimizes the sum of squared distances between the corresponding vector sets without altering the geometric properties of the ETF structure. The objective is to find the rotation matrix $R$ that solves the following optimization problem:

$$\min_R \|RW_{\mathrm{ETF}} - T\|_F^2 \quad \text{subject to} \quad R^T R = I, \tag{3}$$

where $I$ is the identity matrix, $\|\cdot\|_F$ denotes the Frobenius norm.

This problem has a well-known closed-form solution provided by the Kabsch algorithm (see Appendix A.1 for details). Hence, by denoting the solution of 3.1 as $R^*$, the optimally aligned ETF structure with respect to the data distribution $T$ can be expressed as: $W_{\mathrm{ETF}}^* = R^* W_{\mathrm{ETF}}$. Building upon these prototypes, we formulate a contrastive loss function that encourages input image representations to align closely with their corresponding class prototypes, while being pushed away from the prototypes of other classes. Inspired by the Proxy-NCA loss introduced by Movshovitz-Attias et al. (2017), we define the proposed objective loss $\mathcal{L}_{\mathrm{ETF}}$ as:

$$\mathcal{L}_{\mathrm{ETF}} = \sum_{\boldsymbol{w} \in W_{\mathrm{ETF}}} \left\{ \log \left( \sum_{x \in X_{\boldsymbol{w}}^+} e^{-s(\boldsymbol{z}_x, \boldsymbol{w})} \right) + \frac{1}{N} \log \left( \sum_{x \in X_{\boldsymbol{w}}^-} e^{s(\boldsymbol{z}_x, \boldsymbol{w})} \right) \right\}, \tag{4}$$

where $\boldsymbol{z}_x \in \mathbb{R}^{D \times N}$ denotes the representation of the input data obtained from the image encoder of CLIP; $X_{\boldsymbol{w}}^+$ and $X_{\boldsymbol{w}}^-$ represent the sets of positive and negative samples associated with the class corresponding to prototype $\boldsymbol{w}$; $s(\cdot)$ denotes the cosine similarity between two vectors.

Furthermore, by applying $\mathcal{L}_{\mathrm{ETF}}$ to both generated and real data, we simultaneously enforce the representation of synthetic and real samples to converge together to their corresponding ETF prototypes. Specifically, this encourages the synthesized and real data of the same class to be pulled closer to each other, while being pushed away from the prototypes of other classes.

## 3.2 REGIONAL SUPERVISED CONTRASTIVE LOSS

While encouraging both generated and real data representations to align with their corresponding ETF prototypes ensures representation quality, this approach may be overly restrictive, potentially constraining the natural diversity of learned representations and hindering the model's ability to capture the rich variability inherent in real-world data distributions.

Clustering and sampling data from their neighborhoods has proven effective for diversifying datasets without compromising quality (Zhang et al., 2025; Yang et al., 2025). ProtoAug (Nguyen et al., 2025b) has demonstrated both theoretical and empirical efficiency of clustering and robustness in few-shot learning. However, it directly aligns representations of samples within the same cluster without considering their labels, potentially pairing data from similar regions but different classes.

To address these limitations, we introduce $\mathcal{L}_{\mathrm{RSC}}$, a cluster-sensitive contrastive loss that enables *controllable representation pushing* to enhance both data diversity and cross-class discrimination capability. We modify the Supervised Contrastive loss (Khosla et al., 2020) by incorporating a weighting cluster function $c(x, \bar{x})$ that assigns greater penalties to hard negatives - negative samples

within the same cluster. This controllable pushing mechanism ensures that representations maintain sufficient diversity while preserving class boundaries. The loss is computed as:

$$\mathcal{L}_{\text{RSC}}(x) = - \sum_{p \in P(x)} \log \frac{\exp\big(s(\boldsymbol{z}_x, \boldsymbol{z}_p)/\tau\big)}{\sum_{\bar{x} \in \mathcal{D} \setminus \{x\}} c(x, \bar{x}) \, f\big(\exp(s(\boldsymbol{z}_x, \boldsymbol{z}_{\bar{x}})/\tau)\big)}, \tag{5}$$

where the weighting regional function is:

$$c(x, \bar{x}) = \begin{cases} \sigma + \beta \cdot s(\boldsymbol{z}_x, \boldsymbol{z}_{\bar{x}}) & \text{if } y_x \neq y_{\bar{x}} \wedge r_x = r_{\bar{x}}, \\ \sigma & \text{otherwise.} \end{cases}$$

Here, $f$ is the CLIP image encoder, $x$ is the sample from the dataset $\mathcal{D}$, $P(x)$ is the set of positive samples for $x$, and $\bar{x}$ iterates over all other samples. $s(\cdot)$ denotes the cosine similarity between two vectors. The terms $y_x$ and $r_x$ denote the label and the region index for sample $x$. Finally, $\tau$ is the temperature and $\sigma$ and $\beta$ are the weighting hyperparameters. Next, we provide the gradient analysis of the RSC loss function in Theorem 3.1, showcasing the its controllability in enhancing the diversity of data features. A dedicated proof can be found in Appendix B.

**Theorem 3.1.** *Assuming the feature vectors are normalized, the gradient of the loss function $\mathcal{L}_{RSC}(x)$ with respect to the feature vector $\boldsymbol{z}_x$ is given by:*

$$\nabla_{\boldsymbol{z}_x} \mathcal{L}_{RSC}(x) = -\frac{1}{\tau} \sum_{p \in P(x)} \boldsymbol{z}_p + |P(x)| \cdot \frac{\sum_{\bar{x} \in \mathcal{D} \setminus \{x\}} W_{\bar{x}} \cdot \boldsymbol{z}_{\bar{x}}}{\sum_{\bar{x} \in \mathcal{D} \setminus \{x\}} c(x, \bar{x}) \cdot f(\exp(s(\boldsymbol{z}_x, \boldsymbol{z}_{\bar{x}})/\tau))},$$

*where the weight $W_{\bar{x}}$ for each negative sample $\bar{x}$ is defined as:*

$$W_{\bar{x}} = \mathbb{I}(y_x \neq y_{\bar{x}} \wedge r_x = r_{\bar{x}}) \beta f(E_{x\bar{x}}) + \frac{1}{\tau} c(x, \bar{x}) E_{x\bar{x}} f'(E_{x\bar{x}})$$

*and $E_{x\bar{x}} = \exp(s(\boldsymbol{z}_x, \boldsymbol{z}_{\bar{x}})/\tau)$.*

Theorem 3.1 shows that the term:

$$|P(x)| \cdot \frac{\sum_{\bar{x} \in \mathcal{D} \setminus \{x\}} W_{\bar{x}} \, \boldsymbol{z}_{\bar{x}}}{\sum_{\bar{x} \in \mathcal{D} \setminus \{x\}} c(x, \bar{x}) \, f\big(\exp(s(\boldsymbol{z}_x, \boldsymbol{z}_{\bar{x}})/\tau)\big)}$$

acts as a *controllable 'push' gradient* pushes the input data feature $\boldsymbol{z}_x$ away from its negative samples. Because the regional weighting function $c(x, \bar{x})$ modulates the contribution of 'push' gradient for each negative sample, the proportional 'push' gradient assigned to hard-negatives increases while the relative 'push' gradient of others decrease. Consequently, $\boldsymbol{z}_x$ is encouraged to move further away from the directions of hard negatives in the feature space. Specifically, in the few-shot setting, where synthetic samples significantly outnumber real data, this mechanism becomes especially important. Since the quality of synthetic data depends heavily on the generator, features from synthetic data could resemble features of other classes more than those of their own class. As a result, our proposed loss function $\mathcal{L}_{\text{RSC}}$ can decouple numerous pairs of analogous synthetic data but from different classes, resulting in better inter-class distinctiveness of the synthetic data representations, hence increasing the representation diversity.

### 3.3 TRAINING PROCEDURES

The overall training procedure is summarized in the following steps. The detailed algorithm can be found in Appendix C.

1. **Synthesizing Procedure**: Following DataDream (Kim et al., 2024), we fine-tune Stable Diffusion (Rombach et al., 2022) with LoRA (Hu et al., 2022) on the few-shot dataset $\mathcal{D}^{\text{fs}}$. The resulting generator is then employed to generate synthetic data, forming the synthesized dataset $\mathcal{D}^{\text{synth}}$.

2. **Initialization**: First, we employ large language models (LLMs) to generate detailed descriptions for each label. These descriptions are then passed through the pretrained CLIP text-encoder to obtain class-specific representations of the dataset $\mathcal{D}^{\text{fs}}$ for the following alignment procedure. Next, the initial $W_{\text{ETF}}$ class prototypes are randomly initialized according to the ETF structure (Definition 2.1).

| Method | IN | CAL | DTD | EuSAT | AirC | Pets | Cars | SUN | Food | FLO | Avg |
|---|---|---|---|---|---|---|---|---|---|---|---|
| CLIP (Radford et al., 2021) | 70.2 | 96.1 | 46.1 | 38.1 | 23.8 | 91.0 | 63.1 | 72.2 | 85.1 | 71.8 | 64.1 |
| CaFo (Zhang et al., 2023) | 73.9 | 96.9 | 72.5 | 86.7 | 47.4 | 94.9 | 85.7 | 76.9 | 87.6 | 97.8 | 82.0 |
| IsSynth(He et al., 2023) | 73.9 | 97.4 | 75.1 | 93.9 | 64.8 | 92.1 | 88.5 | 77.7 | 86.0 | 99.0 | 84.8 |
| DISEF (da Costa et al., 2023) | 73.8 | 97.0 | 74.3 | 94.0 | 64.3 | 92.6 | 87.9 | 77.6 | 86.2 | 99.0 | 84.7 |
| DataDream$_{cls}$ (Kim et al., 2024) | 73.8 | 97.6 | 73.1 | 93.8 | 68.3 | 94.5 | 91.2 | 77.5 | 87.5 | 99.4 | 85.7 |
| DataDream$_{dset}$ (Kim et al., 2024) | 74.1 | 96.9 | 74.1 | 93.4 | 72.3 | 94.8 | 92.4 | 77.5 | 87.6 | 99.4 | 86.3 |
| ProtoAug (Nguyen et al., 2025b) | 73.8 | 97.3 | 74.5 | 94.7 | 74.3 | 94.6 | 93.1 | 77.7 | **90.4** | 99.3 | 87.0 |
| ImagineFSL (Haoyuan et al., 2025) | **75.2** | 97.9 | **78.0** | 95.0 | 74.1 | **95.4** | 92.9 | **78.8** | 88.3 | **99.7** | 87.5 |
| SyNC (ours) | 73.8 | **97.9** | 75.7 | **95.0** | **80.3** | 95.2 | **93.7** | 77.7 | 90.4 | 99.4 | **87.9** |

Table 1: Few-shot image classification accuracy (%) using CLIP ViT-B/16 with 16 real shots per class. Baseline methods are evaluated across 3 random seeds, while ProtoAug and SyNC are fixed to run for seed 0. **Bold** indicates best performance and underlined indicates second-best performance.

3. **Model Training**: The classifier is fine-tuned on the union of the real dataset $\mathcal{D}^{fs}$ and the synthetic dataset $\mathcal{D}^{synth}$ under our unified losses. In particular, the initial class prototypes $W_{ETF}$ align with the class-specific representations $T$ through the Kabsch rotation to yield the optimal class prototypes $W_{ETF}^*$. Subsequently, $\mathcal{L}_{ETF}$ and $\mathcal{L}_{RSC}$ are computed as defined in Equations 4 and 5.

Finally, the CLIP model is trained with the final loss function defined as follows.

$$\mathcal{L} = \lambda \mathcal{L}_{real} + \mathcal{L}_{syn} + \lambda_1 \mathcal{L}_{ETF} + \lambda_2 \mathcal{L}_{RSC}, \tag{6}$$

where hyperparameters $\lambda$, $\lambda_1$, and $\lambda_2$ are introduced to control the influence of the cross-entropy losses $\mathcal{L}_{real}$ and $\mathcal{L}_{syn}$, and the two proposed losses $\mathcal{L}_{ETF}$ and $\mathcal{L}_{RSC}$, respectively, in the optimization process.

## 4 EXPERIMENTS

### 4.1 EXPERIMENTAL SETUP

**Datasets.** We evaluate our method on 10 benchmark few-shot image classification datasets: Caltech101 (Li et al., 2022) and ImageNet (Russakovsky et al., 2015) for general object recognition, FGVC Aircraft (Maji et al., 2013) and Stanford Cars (Krause et al., 2013) for fine-grained recognition, Food101 (Bossard et al., 2014) for food classification, EuroSAT (Helber et al., 2019) for satellite imagery, Oxford Pets (Parkhi et al., 2012) for animal breeds, DTD (Cimpoi et al., 2014) for texture recognition, SUN397 (Xiao et al., 2010) for scene understanding, and Flowers102 (Nilsback & Zisserman, 2008) for flower species classification.

**Baselines.** We compare our method with existing state-of-the-art methods in Few-shot Image classification with synthetic data: CaFo (Zhang et al., 2023), IsSynth (He et al., 2023), DISEF (da Costa et al., 2023), DataDream (Kim et al., 2024), ProtoAug (Nguyen et al., 2025b), and ImagineFSL (Haoyuan et al., 2025). All of the results of the baseline methods are obtained from the ProtoAug (Nguyen et al., 2025b) and ImagineFSL (Haoyuan et al., 2025) papers.

**Experiment Settings.** We fine-tune CLIP ViT-B/16 image and text encoders using LoRA (Hu et al., 2022). Following baseline protocols, we use Stable Diffusion (SD) v2.1 (Rombach et al., 2022) with guidance scale 2.0 and generated 500 unfiltered images per class. This setup matches all baselines except CaFo and ImagineFSL, which use SD v3 (Esser et al., 2024) to generate 300 filtered images per class plus 300K samples for self-supervised learning. Our approach requires only CLIP encoder fine-tuning, avoiding the additional complexity of self-supervised and two-branch training used by CaFo and ImagineFSL.

The hyperparameters to be tuned are the hyperparameters $\lambda_1, \lambda_2$ to control the $\mathcal{L}_{ETF}$ and $\mathcal{L}_{RSC}$, the learning rates, and the weight decay. The hyperparameter $\lambda$ is chosen at 4 for all datasets except Stanford Cars, where we set it to 1. This choice and the choice of the number of clusters as twice the number of classes are consistent with the choices in ProtoAug (Nguyen et al., 2025b). For

| Methods | Aircraft | | | | Cars | | | |
|---|---|---|---|---|---|---|---|---|
| | 1-shot | 4-shot | 8-shot | 16-shot | 1-shot | 4-shot | 8-shot | 16-shot |
| DataDream (Kim et al., 2024) | 31.1 | 38.3 | 54.6 | 72.3 | 72.9 | 82.6 | 87.4 | 92.4 |
| ProtoAug (Nguyen et al., 2025b) | 25.3 | _51.6_ | _63.9_ | _74.3_ | 72.1 | 86.9 | _91.3_ | _93.1_ |
| ImagineFSL (Haoyuan et al., 2025) | **34.0** | 47.1 | 59.0 | 74.1 | **82.8** | _87.0_ | 89.5 | 92.9 |
| SyNC (ours) | _32.3_ | **53.7** | **68.3** | **80.3** | _75.3_ | **89.0** | **91.8** | **93.7** |

Table 2: Few-shot image classification accuracy (%) with CLIP ViT-B/16 on fine-grained datasets FGVC Aircraft (Maji et al., 2013) and Stanford Cars (Krause et al., 2013) under 1-shot, 4-shot, 8-shot, and 16-shot settings.

the RSC loss function, the parameter $\sigma$, $\beta$, and the temperature $\tau$ are chosen to be 1, 0.25, and 0.07 in all experiments. More details of the hyperparameter settings and tuning process can be found in Appendix D. We compute the loss using CLIP image encoder final-layer representations. However, for the ImageNet dataset, the number of classes (1000) exceeds the dimensionality of CLIP final-layer representations (512). This conflicts with the requirements of the ETF structure defined in Definition. 2.1 (feature dimension $D \geq N - 1$ (number of classes)); so we extract data representations from the penultimate layer of the CLIP model (i.e., prior to the linear projection layer), where the feature dimensionality equals 3072.

## 4.2 MAIN RESULTS

### 4.2.1 GENERAL FEW-SHOT CLASSIFICATION

Table 1 presents the results on 10 benchmarks, comparing our method with original CLIP and recent few-shot approaches. Our approach achieves a new state-of-the-art with an average accuracy of 87.9%, consistently outperforming previous work. The improvement is particularly pronounced on fine-grained datasets, where our method achieves 80.3% on Aircraft and 93.7% on Cars, surpassing ImagineFSL by 6.2% and 0.8%, respectively. These results highlight the robustness of our framework in handling subtle inter-class differences, a setting where many prior approaches struggle. Beyond fine-grained recognition, our method also delivers strong performance in diverse domains, including EuroSAT (95.0%), Pets (95.2%) and Food (90.4%), confirming its broad generalization ability. Overall, these findings demonstrate that our approach not only achieves the best average performance but also provides significant gains in fine-grained scenarios, a key indicator of effective few-shot adaptation. We investigate this phenomenon in more detail in the next subsection.

### 4.2.2 FINE-GRAINED CLASSIFICATION

Table 2 provides detailed comparisons with DataDream, ProtoAug, and ImagineFSL across 1-shot, 4-shot, 8-shot, and 16-shot settings on fine-grained benchmarks, where subtle inter-class variations make classification particularly demanding. Our method achieves substantial improvements on Aircraft: 2.1% in 4-shot, 4.4% in 8-shot, and 6.0% in 16-shot compared to ImagineFSL. On Cars, we establish new state-of-the-art performance with consistent improvements among all settings except the extreme 1-shot scenario. Notably, performance gains become more pronounced with increasing shots, indicating our method more effectively leverages additional training examples to refine class boundaries. This validates our hypothesis that ETF-based prototype alignment enhances the model's ability to capture subtle discriminative features essential for fine-grained recognition.

## 4.3 ABLATION STUDIES

To better understand the contribution of each component in our loss design, we perform ablation studies on all ten datasets. The results in Table 3 show that integrating either ETF or RSC consistently improves performance. Adding ETF loss alone provides moderate gains, particularly achieves the best results on Aircraft, Food and Pets dataset. Meanwhile, the RSC loss alone consistently improves performance in all datasets. When both components are combined, we observe the strongest results across almost all datasets, achieving the best results on the remaining seven. This confirms that ETF and RSC losses are complementary, and their synergy drives robust improvements in fine-

| ETF | RSC | IN | CAL | DTD | EuSAT | AirC | Pets | Cars | SUN | Food | FLO |
|-----|-----|------|------|------|-------|------|------|------|------|------|------|
|     |     | 73.7 | 96.9 | 74.1 | 93.5 | 72.5 | 93.8 | 92.6 | 77.6 | 87.9 | 99.3 |
|     | ✓   | 73.8 | 97.4 | 74.2 | 93.9 | 75.0 | 94.9 | 92.9 | 77.7 | 90.1 | 99.4 |
| ✓   |     | 73.8 | 97.5 | 74.3 | 94.5 | **80.3** | **95.2** | 93.2 | 77.7 | **90.4** | 99.3 |
| ✓   | ✓   | **73.8** | **97.9** | **75.7** | **95.0** | 76.8 | 94.9 | **93.7** | 77.7 | 90.2 | **99.4** |

Table 3: Ablation of the loss function regularization components on all datasets.

tuning performance. In addition, we conduct an ablation study on the Kabsch algorithm, validating its effectiveness in the Appendix A.2.

### 4.4 ANALYSIS

In this section, we analyze the effects of our regularization terms on quality and diversity of representations both quantitatively and qualitatively. Quantitatively, we measure the quality and diversity based on alignment and uniformity metrics, respectively, as proposed in Wang & Isola (2020). We have done this analysis by analyzing these metrics of trained models with and without our loss components on the test sets of each dataset. The results are shown in Figure 2. Qualitatively, we visualize the PCA components of image representation in Appendix E.

Let the test set of data points be $\mathcal{D}^{\text{test}} = \{x_i\}_{i=1}^{M}$, Let $P_{\text{test}}$ be the set of all index pairs $(i,j)$ where samples $x_i$ and $x_j$ belong to the same class. The alignment and uniformity metrics can be written formally as follows.

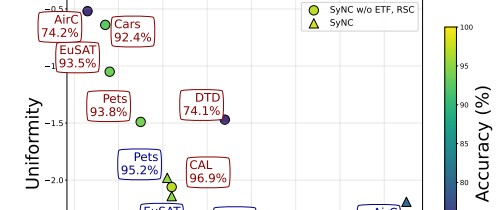

Figure 2: Visualization of the trade-off between alignment, uniformity, and accuracy over multiple datasets. Triangles (△) denote "SyNC" and circles (○) denote "SyNC without ETF, RSC", with color indicating accuracy and annotations highlighting dataset-specific results.

$$\text{Alignment}(f; \alpha) = \frac{1}{|P_{\text{test}}|} \sum_{(i,j) \in P_{\text{test}}} \|f(x_i) - f(x_j)\|_2^\alpha, \tag{7}$$

$$\text{Uniformity}(f; t) = \log \left( \frac{1}{M^2} \sum_{i=1}^{M} \sum_{j=1}^{M} e^{-t\|f(x_i) - f(x_j)\|_2^2} \right). \tag{8}$$

Here, we choose $f$ as the CLIP trained image encoder, and hyperparameter $\alpha$ and $t$ both equal to 2. Figure 2 illustrates how our additional loss components modulate the alignment-diversity trade-off in image representations. Across most datasets, our proposed method substantially enhances feature diversity (significantly reducing uniformity metrics) without compromising quality much (modest increases in alignment metrics). The notable performance gains observed in the DTD and FGVC aircraft datasets suggest that, in few-shot learning scenario, prioritizing feature diversity over perfect alignment can lead to improved classification accuracy.

## 5 CONCLUSION

In this paper, we propose SyNC, a novel training framework that enhances few-shot fine-tuning performance using synthetic data. Experimental results demonstrate its superiority over state-of-the-art methods, especially on fine-grained datasets. Our analysis reveals a strong correlation between the fidelity and diversity of the synthetic data and the overall performance of the model. Although our method shows limitations on large-scale general datasets such as ImageNet, we believe that this can be addressed by incorporating detailed generated captions during data synthesis to improve sample quality. Future work could explore other Neural Collapse variants, such as ETF structures for imbalanced data or improvements to Proxy-NCA loss functions.

## ETHICS STATEMENT

This work aims to enhance few-shot model performance through improved synthetic data utilization, which holds promise for addressing data scarcity scenarios in machine learning applications. By reducing reliance on expensive human annotation, our approach can potentially democratize access to high-performance models and lower barriers for organizations with limited labeling resources. However, since our methods rely on synthetic data generation, they inherit concerns associated with generative models, including potential copyright infringement from training data memorization and amplification of biases present in the underlying datasets. We acknowledge these risks and recommend the careful evaluation of synthetic data sources and the implementation of appropriate bias mitigation strategies when deploying such systems.

## REPRODUCIBILITY STATEMENT

In the paper, to ensure reproducibility, we have described the detailed settings in Section 4.1, including the datasets, baselines, and experimental details. All datasets used in experimental results are publicly available. We further provide detailed algorithm and description of input prompts for closed-source GPT-4 models in Appendix C. We also provide the hyperparameter settings, analysis and searching process in Appendix D. The full code and implementation will be released upon acceptance.

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

# A  KABSCH ALGORITHM

## A.1  IMPLEMENTATION OF KABSCH ALGORITHM IN NEURAL COLLAPSE CONSTRUCTION

Let's recall $T \in \mathbb{R}^{D \times N}$ is the matrix formed from class-specific feature vectors. $W_{\text{ETF}} \in \mathbb{R}^{D \times N}$ is a matrix constructed from initial class prototypes. The Kabsch algorithm proceeds as follows:

1. **Compute the Covariance Matrix:** First, we compute the covariance matrix $H \in \mathbb{R}^{D \times D}$ between the two sets of vectors:
$$H = W_{\text{ETF}} T^T. \tag{9}$$

2. **Singular Value Decomposition (SVD):** Next, we perform Singular Value Decomposition on the covariance matrix $H$:
$$H = U \Sigma V^T, \tag{10}$$
where $U$ and $V$ are orthogonal matrices and $\Sigma$ is a diagonal matrix of singular values.

3. **Calculate the Optimal Rotation:** The optimal rotation matrix $R$ that solves the objective function is then computed as:
$$R = V U^T. \tag{11}$$

A correction step is performed to ensure $R$ represents a pure rotation and not a reflection by checking the determinant of the resulting matrix. If $\det(R) = -1$, the sign of the last column of $V$ is flipped before re-computing $R$.

Finally, the aligned classifier weights, $W_{\text{ETF}}^*$, are obtained by applying the optimal rotation to the initial ETF weights:
$$W_{\text{ETF}}^* = R W_{\text{ETF}}. \tag{12}$$

This alignment procedure initializes our classifier in a semantically meaningful orientation within the CLIP feature space, directly mapping the initial decision boundaries to the geometry of the target class embeddings.

## A.2  ABLATION STUDY OF KABSCH ALGORITHM

Table 4 presents the effectiveness of applying the Kabsch algorithm to the performance of the model. It can be seen that by improving the quality of ETF initialization with Kabsch algorithm, the performance shows a consistent improvement over all datasets.

| Method | IN | CAL | DTD | EuSAT | AirC | Pets | Cars | SUN | Food | FLO |
|---|---|---|---|---|---|---|---|---|---|---|
| No Kabsch | 73.7 | 97.7 | 75.6 | 94.5 | 76.8 | 95.0 | 93.1 | 77.7 | 90.2 | 99.3 |
| Kabsch | **73.8** | **97.9** | **75.7** | **95.0** | **80.3** | **95.2** | **93.7** | 77.7 | **90.4** | **99.4** |

Table 4: Comparison of performance "with" and "without" the Kabsch algorithm across datasets. The best results are shown in **bold**.

# B  PROOF OF THEOREM 3.1

In this section, we provide a comprehensive derivation of the gradient expressions for our proposed loss function $\mathcal{L}_{\text{RSC}}$ (Eq. 5), with respect to an input data representation $z_x$.

The loss for a sample $x$ is defined as:
$$\mathcal{L}_{\text{RSC}}(x) = - \sum_{p \in P(x)} \log \frac{\exp(s(z_x, z_p)/\tau)}{\sum_{\bar{x} \in \mathcal{D} \setminus \{x\}} c(x, \bar{x}) \cdot f(\exp(s(z_x, z_{\bar{x}})/\tau))},$$

where the weighting regional function is:
$$c(x, \bar{x}) = \begin{cases} \sigma + \beta \cdot s(z_x, z_{\bar{x}}) & \text{if } y_x \neq y_{\bar{x}} \wedge r_x = r_{\bar{x}}, \\ \sigma & \text{otherwise.} \end{cases}$$

Here, we rewrite the $\mathcal{L}_{\text{RSC}}$ loss as:

$$\mathcal{L}_{\text{RSC}}(x) = -\sum_{p \in P(x)} \left[ \log(\exp(s(\boldsymbol{z}_x, \boldsymbol{z}_p)/\tau)) - \log\left( \sum_{\bar{x} \in \mathcal{D} \setminus \{x\}} c(x, \bar{x}) \cdot f\left( \exp\left( \frac{s(\boldsymbol{z}_x, \boldsymbol{z}_{\bar{x}})}{\tau} \right) \right) \right) \right] \tag{13}$$

$$= \sum_{p \in P(x)} \left[ -\frac{s(\boldsymbol{z}_x, \boldsymbol{z}_p)}{\tau} + \log\left( \sum_{\bar{x} \in \mathcal{D} \setminus \{x\}} c(x, \bar{x}) \cdot f\left( \exp\left( \frac{s(\boldsymbol{z}_x, \boldsymbol{z}_{\bar{x}})}{\tau} \right) \right) \right) \right] \tag{14}$$

## B.1 GRADIENT CALCULATION

We now compute the gradient of equation 14 with respect to $\boldsymbol{z}_x$. Let the denominator sum be denoted by $D$:

$$D = \sum_{\bar{x} \in \mathcal{D} \setminus \{x\}} c(x, \bar{x}) \cdot f\left( \exp\left( \frac{s(\boldsymbol{z}_x, \boldsymbol{z}_{\bar{x}})}{\tau} \right) \right).$$

The gradient calculation can be split into two parts:

$$\nabla_{\boldsymbol{z}_x} \mathcal{L}_{\text{RSC}}(x) = \sum_{p \in P(x)} \left[ \nabla_{\boldsymbol{z}_x} \left( -\frac{s(\boldsymbol{z}_x, \boldsymbol{z}_p)}{\tau} \right) + \nabla_{\boldsymbol{z}_x} (\log(D)) \right]$$

*Assumption.* We assume that the representation are normalized during training process, so the norm is equal to 1. So the cosine similarity between two representations could be simplified to the dot product function.

**Lemma B.1** (Gradient of the Positive Term). *The gradient of the term corresponding to positive samples is:*

$$\nabla_{\boldsymbol{z}_x} \left( -\frac{s(\boldsymbol{z}_x, \boldsymbol{z}_p)}{\tau} \right) = -\frac{1}{\tau} \boldsymbol{z}_p$$

*Proof.* This follows directly from the linearity of the gradient and the assumption above:

$$\nabla_{\boldsymbol{z}_x} \left( -\frac{s(\boldsymbol{z}_x, \boldsymbol{z}_p)}{\tau} \right) = -\frac{1}{\tau} \nabla_{\boldsymbol{z}_x} s(\boldsymbol{z}_x, \boldsymbol{z}_p) = -\frac{1}{\tau} \boldsymbol{z}_p$$

$\square$

**Lemma B.2** (Gradient of the Negative Term). *The gradient of the term corresponding to the sum over negative samples is:*

$$\nabla_{\boldsymbol{z}_x} \log(D) = \frac{1}{D} \sum_{\bar{x} \in \mathcal{D} \setminus \{x\}} W_{\bar{x}} \cdot \boldsymbol{z}_{\bar{x}}$$

*where $W_{\bar{x}} = \mathbb{I}(y_x \neq y_{\bar{x}} \wedge r_x = r_{\bar{x}})\beta f(E_{x\bar{x}}) + \frac{1}{\tau} c(x, \bar{x}) E_{x\bar{x}} f'(E_{x\bar{x}})$, and $E_{x\bar{x}} = \exp(s(\boldsymbol{z}_x, \boldsymbol{z}_{\bar{x}})/\tau)$.*

*Proof.* Using the chain rule, we have $\nabla_{\boldsymbol{z}_x} \log(D) = \frac{1}{D} \nabla_{\boldsymbol{z}_x} D$. We find $\nabla_{\boldsymbol{z}_x} D$ by differentiating term-wise and applying the product rule $\nabla(uv) = (\nabla u)v + u(\nabla v)$:

$$\nabla_{\boldsymbol{z}_x} [c(x, \bar{x}) \cdot f(E_{x\bar{x}})] = (\nabla_{\boldsymbol{z}_x} c(x, \bar{x})) f(E_{x\bar{x}}) + c(x, \bar{x}) (\nabla_{\boldsymbol{z}_x} f(E_{x\bar{x}}))$$

The gradients of the individual components are:

1. $\nabla_{\boldsymbol{z}_x} c(x, \bar{x}) = \mathbb{I}(y_x \neq y_{\bar{x}} \wedge r_x = r_{\bar{x}})\beta \nabla_{\boldsymbol{z}_x} s(\boldsymbol{z}_x, \boldsymbol{z}_{\bar{x}}) = \mathbb{I}(y_x \neq y_{\bar{x}} \wedge r_x = r_{\bar{x}})\beta \boldsymbol{z}_{\bar{x}}$

2. $\nabla_{\boldsymbol{z}_x} f(E_{x\bar{x}})$ requires the chain rule:

$$\nabla_{\boldsymbol{z}_x} f(E_{x\bar{x}}) = \frac{df}{dE_{x\bar{x}}} \cdot \frac{dE_{x\bar{x}}}{ds(\boldsymbol{z}_x, \boldsymbol{z}_{\bar{x}})} \cdot \nabla_{\boldsymbol{z}_x} s(\boldsymbol{z}_x, \boldsymbol{z}_{\bar{x}})$$

$$= f'(E_{x\bar{x}}) \cdot \frac{1}{\tau} \exp\left( \frac{s(\boldsymbol{z}_x, \boldsymbol{z}_{\bar{x}})}{\tau} \right) \cdot \boldsymbol{z}_{\bar{x}}$$

$$= \frac{1}{\tau} f'(E_{x\bar{x}}) E_{x\bar{x}} \boldsymbol{z}_{\bar{x}}$$

Combining these results and factoring out the common vector $z_{\bar{x}}$ yields:

$$\nabla_{z_x}\left[c \cdot f\right] = \left[\mathbb{I}(y_x \neq y_{\bar{x}} \wedge r_x = r_{\bar{x}})\beta f(E_{x\bar{x}})\right] z_{\bar{x}} + \left[c(x, \bar{x})\frac{1}{\tau}f'(E_{x\bar{x}})E_{x\bar{x}}\right] z_{\bar{x}}$$

$$= \underbrace{\left(\mathbb{I}(y_x \neq y_{\bar{x}} \wedge r_x = r_{\bar{x}})\beta f(E_{x\bar{x}}) + \frac{1}{\tau}c(x, \bar{x})E_{x\bar{x}}f'(E_{x\bar{x}})\right)}_{W_{\bar{x}}} z_{\bar{x}}$$

Summing over all $\bar{x}$ and dividing by $D$ completes the proof. $\qquad\square$

## B.2 FINAL GRADIENT EXPRESSION

By combining the results from the above lemmas, we arrive at the final expression for the gradient.

The gradient of the loss function $\mathcal{L}_{\text{RSC}}(x)$ with respect to the feature vector $z_x$ is given by:

$$\nabla_{z_x}\mathcal{L}_{\text{RSC}}(x) = -\frac{1}{\tau}\sum_{p \in P(x)} z_p + |P(x)| \cdot \frac{\sum_{\bar{x} \in \mathcal{D}\setminus\{x\}} W_{\bar{x}} \cdot z_{\bar{x}}}{\sum_{\bar{x} \in \mathcal{D}\setminus\{x\}} c(x, \bar{x}) \cdot f(\exp(s(z_x, z_{\bar{x}})/\tau))}$$

where the weight $W_{\bar{x}}$ for each negative sample $\bar{x}$ is defined as:

$$W_{\bar{x}} = \mathbb{I}(y_x \neq y_{\bar{x}} \wedge r_x = r_{\bar{x}})\beta f(E_{x\bar{x}}) + \frac{1}{\tau}c(x, \bar{x})E_{x\bar{x}}f'(E_{x\bar{x}})$$

and $E_{x\bar{x}} = \exp(s(z_x, z_{\bar{x}})/\tau)$.

*Proof.* We assemble the gradient by summing the components for each $p \in P(x)$:

$$\nabla_{z_x}\mathcal{L}_{\text{RSC}}(x) = \sum_{p \in P(x)}\left[-\frac{1}{\tau}z_p + \frac{\sum_{\bar{x}} W_{\bar{x}}z_{\bar{x}}}{D}\right]$$

Since the second term is constant with respect to the summation over $p$, we can rewrite it as:

$$\nabla_{z_x}\mathcal{L}_{\text{RSC}}(x) = \left(\sum_{p \in P(x)} -\frac{1}{\tau}z_p\right) + |P(x)|\left(\frac{\sum_{\bar{x}} W_{\bar{x}}z_{\bar{x}}}{D}\right)$$

This yields the final expression as stated in the theorem. $\qquad\square$

# C DETAILED ALGORITHM

## C.1 ALGORITHM PIPELINE

---
**Algorithm 1** Training procedure

---
**Input**:
   Real dataset: $\mathcal{D}^{\text{fs}}$.
   Test dataset: $\mathcal{D}^{\text{test}}$.
   Pretrained generator model: $\mathcal{G}$,
   Learning rate schedule: $\eta$

1: Fine-tuning generator $\mathcal{G}$ by real dataset $\mathcal{D}^{\text{fs}}$ with LoRA
2: Generate $T$ synthetic images from generator $\mathcal{G}$, to construct the synthesized dataset $\mathcal{D}^{\text{synth}}$.
3: Let $\mathcal{D} = \mathcal{D}^{\text{fs}} \cup \mathcal{D}^{\text{synth}}$.
4: Generate $Des$, the set of descriptions for all classes, generated by LLMs.
5: Initialization random ETF structure: $W_{\text{ETF}}$
6: Construct optimal $W_{\text{ETF}}^*$ from $W_{\text{ETF}}$ and $Des$ using Kabsch algorithm.
7: Use K-means clustering on both real and synthetic images to obtain regions.
8: **for** batch in batches($\mathcal{D}$) **do**
9:    Compute $\mathcal{L}_{\text{ETF}}$ as in Eq.4
10:   Compute $\mathcal{L}_{\text{RSC}}$ as in Eq. 5
11:   Fine-tuning models with loss function in Eq. 6
12: **end for**

---

| Hyperparameter | Value |
|---|---|
| Batch size | 128 |
| Epochs | 50 |
| Optimizer | AdamW |
| Learning rate | $\{2e-4, 1e-4, 1e-5, 1e-6\}$ |
| Weight decay | $\{1e-3, 5e-4, 1e-4\}$ |
| LR schedule | Cosine schedule |
| Augmentation | Similar to all baselines |
| ETF $\lambda_1$ | $\{0.05, 0.2, 0.5, 1\}$ |
| RSC $\lambda_2$ | $\{0, 0.05, 0.1\}$ |
| RSC $\sigma$ | 1 |
| RSC $\beta$ | 0.25 |
| RSC $\tau$ | 0.07 |
| Number of clusters $K$ | Twice as the number of classes |
| LoRA rank | 16 |
| LoRA weight | 32 |
| LoRA dropout | 0.1 |

Table 5: Hyperparameters setting for tuning process

| $\lambda_1$ | CAL | DTD | EuSAT | Pets | Cars | Food |
|---|---|---|---|---|---|---|
| 0.05 | 97.2 | **74.3** | 94.6 | 94.9 | 92.9 | 90.1 |
| 0.2 | 97.2 | 74.2 | **94.7** | 94.9 | 92.9 | 90.3 |
| 0.5 | 97.4 | 72.8 | 94.4 | **95.2** | **93.2** | **90.4** |
| 1 | **97.5** | 73.7 | 93.9 | 94.6 | **93.2** | 90.1 |

Table 6: Hyperparameter search for $\lambda_1$.

## C.2 DETAILED DESCRIPTION BY GPT-4

We construct comprehensive textual descriptions for every class label in the dataset. We use the following guidance for the GPT-4 as input prompts: Each description must be precisely 77 tokens in length to align with the input constraints of the CLIP text encoder. The descriptions should emphasize distinctive, class-specific attributes that effectively differentiate each label from the others. The final output is presented as a Python dictionary that matches each label to its corresponding description.

## D IMPLEMENTATION DETAILS

### D.1 HYPERPARAMETER SETTINGS

The hyperparameter values for the classifier tuning process are presented in Table 5. The clustering phase was performed with the FAISS library Douze et al. (2024). The main hyperparameters to be tuned include learning rate, weight decay, and the weighting hyperparameters $\lambda_1$ and $\lambda_2$ of ETF and RSC loss components. The detail process of choosing these hyperparameters are presented in the next subsection.

For the generating process, we follow the same settings of DataDream (Kim et al., 2024) and ProtoAug (Nguyen et al., 2025b). We fine-tune the Stable Diffusion version 2.1 with LoRA using the learning rate of $1e-4$. Then, we generate 500 images per class without filtering, setting the guidance scale to be 2.

| $\lambda_2$ | CAL | DTD | EuSAT | Pets | Cars | Food |
|---|---|---|---|---|---|---|
| 0.05 | **97.9** | **75.7** | 94.7 | **94.9** | 93.4 | **90.4** |
| 0.1 | 97.5 | 75.5 | **95.0** | **94.9** | **93.7** | 90.2 |

Table 7: Hyperparameter search for $\lambda_2$.

### D.2 HYPERPARAMETER SEARCH DETAILED PROCESS

We search for the hyperparameter $\lambda_1$ and $\lambda_2$ to control the $\mathcal{L}_{\text{ETF}}$ and $\mathcal{L}_{\text{RSC}}$ using the coordinate ascent. We choose $\lambda_1$ after some initial experiments, we observe that in order to maintain a good ratio between $\mathcal{L}_{\text{ETF}}$ and the cross-entropy loss $\mathcal{L}_{real}$ and $\mathcal{L}_{syn}$, the optimal choice lies in the range of $[0.05, 1]$. So we search for $\lambda_1$ in the set of $\{0.05, 0.2, 0.5, 1\}$. The experimental results for some dataset are presented in Table 6. All of the dataset are chosen with these values, except FGVC Aircraft. This dataset is a well-known challenge in this field, due to the fine-grained structure and the weak performance of Stable Diffusion in this aircraft domain. In this special case, we found that increasing the value of $\lambda_1$ from 1 up to 15 still increase the performance before saturation. So we choose $\lambda_1$ to be 15 in this case.

Then, based on the selected values of $\lambda_1$, we conduct experiments to decide $\lambda_2$. Also after some initial experiments on small dataset, we choose $\lambda_2$ out of 0, 0.05 and 0.1. The choice of 0 is when we observe ETF alone achive the best results in Table 3. The remaining results of this choice is presented in Table 7.

Overall, the results in Table 6 and 7 show that SyNC is robust with logical hyperparameter settings, and once we maintain a good cross-entropy and regularization ratio, SyNC can achieve satisfactory results.

The remaining hyperparameters, including learning rate, weight decay are tuned by grid search with early stopping. For the augmentation process, we observe that because our method has explicitly working to ensure the optimal representation for data, the use of CutMix (Yun et al., 2019) and Mixup (Zhang et al., 2018) augmentation usually decreases performance, with some minor exception. So in order to yield the robust performance, we would recommend not using these augmentation as defaults. The remaining data augmentation methods are used the same as all of the baselines.

## E QUALITATIVE ANALYSIS OF DATA REPRESENTATION

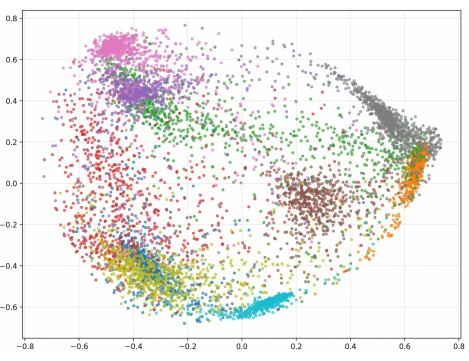
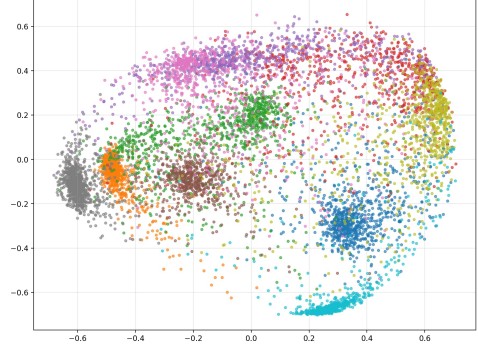

Figure 3: 2D PCA visualization of learned embeddings from the baseline model (SyNC without ETF, RSC) on the EuroSAT test set.

Figure 4: 2D PCA visualization of learned embeddings from the complete SyNC model on the EuroSAT test set.

Figures 3 and 4 compare 2D PCA visualizations of learned embeddings on the EuroSAT test set. Figure 3 depicts the embedding space obtained by "SyNC without ETF and RSC," whereas Figure 4 illustrates the results of the complete "SyNC" model. The comparison clearly demonstrates that

| Methods | AirC | Cars | Food | CAL |
|---|---|---|---|---|
| Real fine-tune | 61.57 | 78.86 | 63.52 | 93.29 |
| IsSynth | 70.94 | 90.82 | 68.77 | 94.54 |
| DISEF | 65.99 | 79.18 | 70.10 | 94.34 |
| DataDream$_{cls}$ | 79.21 | 92.99 | 66.70 | 94.37 |
| DataDream$_{dset}$ | 81.46 | 93.30 | 66.63 | 94.62 |
| ProtoAug | 82.67 | 93.71 | 70.35 | 94.17 |
| SyNC (ours) | **83.48** | **94.44** | **70.69** | **95.22** |

Table 8: Results of different methods on CLIP-ResNet50 fine-tuning.

integrating ETF and RSC significantly enhances both the diversity and fidelity of the learned data representations.

Specifically, the embedding space in Figure 4 exhibits more compact and well-separated clusters, indicating improved intra-class cohesion and inter-class discrimination. This structural clarity contrasts with Figure 3, where the clusters are diffuse and overlapping, reflecting less reliable feature learning. Moreover, the full SyNC model produces representations with smoother boundaries and reduced noise, suggesting higher fidelity and consistency in the learned features. This enhanced representation quality implies that our approach not only preserves fine-grained distinctions among categories but also aligns samples more effectively with their respective class prototypes.

## F  RESULTS ON DIFFERENT ARCHITECTURES

Following DataDream (Kim et al., 2024) and ProtoAug (Nguyen et al., 2025b), we evaluate SyNC's cross-architectural generalization by fine-tuning CLIP-ResNet50 (Radford et al., 2021) on 16-shot settings across four benchmark datasets: FGVC Aircraft, Stanford Cars, Food-101, and Caltech-101. As shown in Table 8, our method consistently outperforms other baselines across all datasets with improvements ranging from 0.3% to 0.8%, demonstrating SyNC's architectural robustness and generalization capability, beyond the primary experimental setup.

## G  DETAILS OF LARGE LANGUAGE MODELS USAGE

We use Large Language Models (LLMs) solely to improve the writing quality of this manuscript, including grammar, sentence structure, and clarity of expression. The LLMs were not involved in research design, data analysis, interpretation of results, or any intellectual content development. All scientific contributions, methodological designs, and conclusions are entirely the work of the human authors. The LLMs served only as writing assistants to polish the presentation of our manuscript.

