# OpenReview forum: "SyNC: Balancing Fidelity and Diversity of Synthetic Data Representations in CLIP-based Few-Shot Learning via Neural Collapse"
_ICLR.cc/2026/Conference — Submitted to ICLR 2026_

### Official Review · Reviewer_YwNC · 2025-10-19

**Soundness:** 2
**Presentation:** 2
**Contribution:** 2
**Rating:** 4
**Confidence:** 4

**Summary:**

This paper proposes SyNC, a new training paradigm for CLIP-based few-shot learning that explicitly balances the fidelity and diversity of synthetic data representations via two complementary loss functions. The authors identify a key trade-off in existing methods, which often optimize for one aspect at the expense of the other.  Extensive experiments on 10 few-shot image classification benchmarks demonstrate that SyNC achieves a new state-of-the-art average accuracy, with particularly significant improvements on fine-grained datasets like FGVC Aircraft and Stanford Cars. Ablation studies and analysis of alignment/uniformity metrics confirm the contribution of each component and the method's ability to better balance fidelity and diversity.

**Strengths:**

1. $\textbf{Originality}$: This paper demonstrates a degree of originality, primarily through the creative combination of established ideas from disparate fields to address a timely and well-defined problem.

2. $\textbf{Quality}$: This paper exhibits good quality in its methodological execution and empirical validation.

3. $\textbf{Clarity}$: This paper is generally well-written and structured, making a technically complex approach accessible.

**Weaknesses:**

1.$\textbf{Dependence on Synthetic Data Quality}$: The entire framework is contingent on the quality of the initial synthetic data generated by the fine-tuned Stable Diffusion model. While the method is designed to be robust, there is no analysis of how its performance degrades with lower-quality generators or noisier synthetic data.

2.$\textbf{Superficial Theoretical Exploration of NC in FSL}$: While the use of Neural Collapse is a strength, the connection remains somewhat applied. A deeper theoretical discussion on why NC is a suitable inductive bias for the few-shot setting, or an analysis of whether NC actually emerges under the proposed losses, would elevate the contribution further.

**Questions:**

1. How sensitive is SyNC to the quality of the synthetic data? For instance, if you were to use a weaker generative model or reduce the number of generated samples per class, how would the performance of SyNC compare to the baselines?

2. In the RSC loss, how is the "region" or cluster membership $r_x$ precisely defined? Is it based on a standard clustering algorithm like K-means? How sensitive is the performance to the choice of the number of clusters (stated as 2N), and was this hyperparameter ablated?

3. The method uses GPT-4 to generate enriched class descriptions. How critical is this step? Would the technique still achieve significant gains using only the naive "a photo of a [CLS]" prompt?

---

> ### Author Response · Authors · 2025-11-21
> **Thank you for your valuable suggestion**
>
> We thank the reviwer for valuable suggestions. We would like to address your remaining concerns below.
>
> ## Weaknesses:
> **W1.* The entire framework is contingent on the quality of the initial synthetic data generated by the fine-tuned Stable Diffusion model. While the method is designed to be robust, there is no analysis of how its performance degrades with lower-quality generators or noisier synthetic data.
> * Answer:
>
> Thanks for the sugggestion. We want to gently remind that none of the design is related to fine-tuning the image generator (Stable Diffusion). We utilized the fine-tuned version solely to be consistent with the baselines. We provides more results on non-fine-tuned generator here.
>
> ||euro|pets|dtd|calt|flow|airc|food|
> |-|-|-|-|-|-|-|-|
> |No finetune|94.3|92.3|74.1|93.9|98.5|71.2|89.4|
> |Finetuned|95.0|95.2|75.7|97.9|99.4|80.3|90.4|
>
> The results shows that while fine-tuning the generator increase the performance, our method can still perform well on synthetic data synthesized by no fine-tuned generator.
>
> **W2.* While the use of Neural Collapse is a strength, the connection remains somewhat applied. A deeper theoretical discussion on why NC is a suitable inductive bias for the few-shot setting, or an analysis of whether NC actually emerges under the proposed losses, would elevate the contribution further.
> * Answer:
>
> We thank the reviewer for having this suggestion. We want to provide the following discussion about a theoretical validation of Neural Collapse in few-shot settings. We will integrate this discussion into the revised paper.
>
> Existing few-shot settings often employ a protype mechanism, where a class prototype is represented by averaging the embeddings of all samples within that class [1]. While these prototypes reflect a  simple  but yet robust inductive bias, these methods may non-representative prototypes does not guarantee the ability to distinguish between analogous classes effectively. Recently, since [4] introduced a novel phenomenon in deep learning models, numerous theoretical discussion about the use of neural collapse on few-shot trasnfer learning have been introduced in [2,3]. They shows that the test error of target tasks and the generalization transfer error from source data to the new sample of the same classes and new classes, are upper bounded by the empirical version of class-distance normalized variance (CDNV) (a simplified version of NC1 assumption of neural collapse). So in order to achieve good performance on few-shot leaning, one could minimize this quantity CDNV, in turn, minimize the variance within features of the given class, while maximizing its distance to other classes. This theoretical analysis serves as a natural theoretical guided motivation of the use of neural collapse as the prototypes training.
>
> From empirical perspective, NC-based frameworks have already been adopted in numerous few-shot classification settings across both computer vision and NLP. For example, [4] utilized NC principles within FSCIL, and [5] applied the GOF (General Orthogonal Frame) to Few-Shot Continual Relation Extraction (FCRE). These results collectively suggest that NC properties can manifest well before reaching the theoretical asymptotic limits.
>
> [1] Snell, Jake, Kevin Swersky, and Richard Zemel. “Prototypical Networks for Few-Shot Learning.” Advances in Neural Information Processing Systems 30 (NeurIPS), 2017.
>
> [2] "On the Role of Neural Collapse in Transfer Learning" Tomer Galanti, András György, Marcus Hutter, ICLR 2022
>
> [3] "Generalization Bounds for Few-Shot Transfer Learning with Pretrained Classifiers", Tomer Galanti, András György, Marcus Hutter
>
> [4] Papyan, Vardan; Han, X. Y.; Donoho, David L. (2020). Prevalence of Neural Collapse during the terminal phase of deep learning training.

---

> > ### Author Response · Authors · 2025-11-21
> >
> > ## Questions:
> > **Q1.* How sensitive is SyNC to the quality of the synthetic data? For instance, if you were to use a weaker generative model or reduce the number of generated samples per class, how would the performance of SyNC compare to the baselines?
> > * Answer: We have provided results for lower quality generator in Weakness 1, here we provide more results on varying the number of synthetic data. We provide the comparison with the strongest baselines of the same image generation setting (ProtoAug) in the table below.
> >
> > |No. synth. samples|**EuSAT**||**DTD**||**AirC**||
> > |-|-|-|-|-|-|-|
> > ||**ProtoAug**|**SyNC**|**ProtoAug**|**SyNC**|**ProtoAug**|**SyNC**|
> > |100|94.2|**94.3**|73.9|**74.0**|69.6|**72.6**|
> > |200|94.5|**94.9**|74.0|**74.5**|71.9|**75.3**|
> > |300|94.4|**94.8**|73.8|**74.7**|73.0|**77.0**|
> > |400|94.4|**94.5**|74.2|**75.2**|73.3|**76.7**|
> >
> > The table shows that our method consistently outperform the baselines while varying the number of synthesized data, confirming the superiority of our method over current state-of-the-art methods.
> >
> > **Q2.* In the RSC loss, how is the "region" or cluster membership
> >  precisely defined? Is it based on a standard clustering algorithm like K-means? How sensitive is the performance to the choice of the number of clusters (stated as 2N), and was this hyperparameter ablated?
> > * Answer: We use Kmeans for computing the RSC loss, described in the Algorithm 1 in Appendix C.  We provide an ablation studies table on the choice of the number of clusters in the following table, with N is the number of classes in each dataset.
> >
> > |number of clusters|euro|pets|dtd|fgvc|
> > |-|-|-|-|-|
> > |1|92.2|94.2|74.0|74.3|
> > |N|92.3|94.4|74.1|75.4|
> > |2N|**95.0**|**95.2**|**75.7**|**80.3**|
> > |4N|93.2|94.2|75.2|76.8|
> >
> > The table validates the choice of number of clusters as 2N is the optimal choice of the number of clusters. Decreasing this number of clusters can make the RSC loss to be ambiguous, leading to matching  wrong representations of different classes. On the other hand, increasing this number too much may lead to  pushing the representation of the same class too far away from each other, negatively affecting the decision boundaries.
> >
> > **Q3.* The method uses GPT-4 to generate enriched class descriptions. How critical is this step? Would the technique still achieve significant gains using only the naive "a photo of a [CLS]" prompt?
> > * Answer:
> >
> > |Ablation on prompts|eurosat|pets|dtd|calt|flow|cars|
> > |-|-|-|-|-|-|-|
> > |a photo of [CLS]|92.3|94.2|73.6|96.2|**99.4**|93.3|
> > |GPT enhanced prompt|**95.0**|**95.2**|**75.7**|**97.9**|**99.4**|**93.7**|
> >
> > We provide an ablation for the use of GPT-4 prompt in initializing the representation of text encoder. The table shows that the generated prompts is necessary to improve the quality of initialized text representations and in turn the initialization of ETF structure. Note that this is only used to improve the performance of ETF not for generating images to ensure a fair comparison with the baselines.

---

> > > ### Comment · Reviewer_YwNC · 2025-11-25
> > >
> > > I appreciate the authors' response. However, upon reading their reply to my question on $\textbf{Q3}$, it seems my original concern may not have been fully conveyed. To clarify, "the technique" refers to the proposed SyNC rather than using GPT-4 prompts.
> > > Moreover, while the authors have discussed the existing Neural Collapse theories in few-shot settings, they do not establish a new theoretical foundation for their innovation.
> > > Therefore, I maintain my original score.

---

### Official Review · Reviewer_HH2D · 2025-10-24

**Soundness:** 2
**Presentation:** 2
**Contribution:** 2
**Rating:** 4
**Confidence:** 3

**Summary:**

This paper introduces SyNC, a new paradigm that explicitly balances "synthetic feature fidelity vs. diversity" during the small-sample fine-tuning phase of CLIP. The key contributions are:

(1) The introduction of Neural Collapse (NC) theory into the small-sample synthetic data scenario, with the proposal of ETF prototype alignment loss ($L_{ETF}$);

(2) The design of regional supervised contrastive loss ($L_{RSC}$), which applies greater force to "misclassified synthetic samples" to enhance inter-class distance.

**Strengths:**

(1) This is application of the Neural Collapse ETF structure to synthetic data training in few-shot learning. By dynamically aligning prototypes using the Kabsch algorithm, it enhances the authenticity of the representations.

(2) Two complementary loss functions are proposed: the ETF loss ($L_{ETF}$) for authenticity and the regional contrastive loss ($L_{RSC}$) for diversity, addressing the limitation of existing methods that favor one over the other.

**Weaknesses:**

(1) The premise of this paper is based on the existence of the NC phenomenon, but the four key assumptions of NC (large batch size, class balance, sufficient training, and real data) are all violated in the small-sample + synthetic regime. The authors use "ETF prototypes" as an inductive bias but do not provide any finite-sample convergence bounds or generalization error reduction theorems.

(2) The premise of the article is that the NC phenomenon exists in CLIP, but the article lacks proof of the existence of the NC phenomenon in CLIP.

(3) The overall idea of the article is very similar to the KDD 2024 paper "Understanding Prompt Tuning for V-L Models Through the Lens of Neural Collapse." [1] The claimed originality of the article seems significantly diminished, and there is insufficient comparison and explanation.

[1] Didi Zhu, Zexi Li, Min Zhang, Junkun Yuan, Jiashuo Liu, Kun Kuang, and Chao Wu. 2024. Neural Collapse Anchored Prompt Tuning for Generalizable Vision-Language Models. In Proceedings of the 30th ACM SIGKDD Conference on Knowledge Discovery and Data Mining (KDD '24). Association for Computing Machinery, New York, NY, USA, 4631–4640. https://doi.org/10.1145/3637528.3671690

**Questions:**

**Problem:**

(1) The premise of this paper relies on the existence of the Neural Collapse (NC) phenomenon; however, the four key assumptions of NC—large batch size, class balance, sufficient training, and real data—are all violated in the small-sample and synthetic regime employed. The authors propose "ETF prototypes" as an inductive bias but do not provide finite-sample convergence bounds or generalization error reduction theorems to support their approach.

(2) The paper assumes the existence of the NC phenomenon in CLIP, but it lacks empirical proof or formal argumentation to substantiate this claim.

(3) The overall idea of the article closely mirrors the KDD 2024 paper "Understanding Prompt Tuning for V-L Models Through the Lens of Neural Collapse" (Zhu et al., 2024). This similarity significantly diminishes the claimed originality of the paper, and there is insufficient comparison and explanation to clarify how the current work differs from the existing literature.

---

> ### Author Response · Authors · 2025-11-21
> **Thank you for your constructive feedback**
>
> We thank the reviewer for thoughtful questions and valuable feedback. We would like to address your concerns below.
>
> **Q1.* The premise of this paper relies on the existence of the Neural Collapse (NC) phenomenon; however, the four key assumptions of NC—large batch size, class balance, sufficient training, and real data—are all violated in the small-sample and synthetic regime employed. The authors propose "ETF prototypes" as an inductive bias but do not provide finite-sample convergence bounds or generalization error reduction theorems to support their approach.
> * Answer:
>
> Thank you for raising this concern. We believe our training configuration does satisfiy whole conditions associated with the emergence of Neural Collapse:
>
> Few classes: We ensure that the number of classes is upper-bounded by the embedding dimension plus one: C ≤ d + 1. (as the condition for the construction of ETF)
>
> Balanced training: The number of samples is equal across all classes, in term of both real and synthetic data. This satisfies the NC assumption of balanced class sizes.
>
> Noise-free labels: The synthetic samples are created using a LoRA-tuned Stable Diffusion model; hence, the samples within each class exhibit highly similar visual characteristics and precise labels. This aligns with the "noise-free labels" assumption under which NC is known to appear.(Identical embeddings should belong to the same class)
>
> Large training batch size: We consistently employ a training batch size of 128, which is larger than (or comparable to) those used in the original NC studies (e.g., [1], where 64 was standard for ResNet experiments).
>
> From the theoretical perspective, some discussion about the use of neural collapse on few-shot trasnfer learning have been introduced in [2,3]. They shows that the test error of target tasks and the generalization transfer error from source data to the new sample of the same classes and new classes, are upper bounded by the empirical version of class-distance normalized variance (CDNV) (a simplified version of NC1 assumptions of neural collapse). So in order to achieve good performance on few-shot leaning, one could minimize this quantity CDNV, in turn, minimize the variance within features of the given class, while maximizing its distance to other classes. This theoretical analysis serves as a natural theoretical guided motivation of the use of neural collapse as the prototypes training in few-shot learning.
>
> Moreover, from experimental perspective, NC-based frameworks have already been adopted in numerous few-shot classification settings across both computer vision and NLP. For example, [4] utilized NC principles within FSCIL, and [5] applied the GOF (General Orthogonal Frame) to Few-Shot Continual Relation Extraction (FCRE). These results collectively suggest that NC properties can manifest well before reaching the theoretical asymptotic limits.
>
> Hence, using ETF prototypes as an inductive bias is both theoretically motivated and empirically supported.
>
> [1] Papyan, V., Han, X. Y., & Donoho, D. L. (2020). Prevalence of neural collapse during the terminal phase of deep learning training. Proceedings of the National Academy of Sciences (PNAS), 117(40), 24652–24663.
>
> [2] Galanti, T., György, A., & Hutter, M. (2022). On the role of neural collapse in transfer learning. International Conference on Learning Representations (ICLR).
>
> [3] Galanti, T., György, A., & Hutter, M. (2023). Generalization bounds for few-shot transfer learning with pretrained classifiers. arXiv preprint arXiv:2212.12532.
>
> [4] Yang, Y., Yuan, H., Li, X., Lin, Z., Torr, P., & Tao, D. (2023). Neural collapse inspired feature‑classifier alignment for few‑shot class‑incremental learning. International Conference on Learning Representations (ICLR).
>
> [5] Pham, T. D., Le, H., Ngo Van, L., Nguyen, N. T. N. D., Dinh, S., & Nguyen, T. H. (2025). Mitigating non‑representative prototypes and representation bias in few‑shot continual relation extraction. Proceedings of the Annual Meeting of the Association for Computational Linguistics (ACL).

---

> ### Author Response · Authors · 2025-11-21
>
> **Q2.* The paper assumes the existence of the NC phenomenon in CLIP, but it lacks empirical proof or formal argumentation to substantiate this claim.
> * Answer:
>
> Thank you for highlighting the need to justify the emergence of the Neural Collapse (NC) phenomenon in CLIP. We appreciate the opportunity to provide additional clarification.
>
> Recently, [6] computed two metrics from the Neural Collapse literature: NC1 and NC2 on CLIP-based models. Intuitively, NC1 and NC2 measure the compactness and separation of clusters, respectively. NC1 approaches zero when the within-class variation of features becomes negligible, while NC2 converges to zero when classifiers achieve maximal and equal margins, corresponding to an ETF structure. These experiments show that the feature distribution of CLIP pre-trained datasets exhibits a strong correlation between NC2 and per-class accuracy. In short, lower NC2 reliably indicates where CLIP is likely to fail, whereas higher NC2 corresponds to stronger discriminability.
>
> [6] Wen, X., Zhao, B., Chen, Y., Pang, J., & Qi, X. (2024). What makes CLIP more robust to long-tailed pre-training data? A controlled study for transferable insights. Advances in Neural Information Processing Systems (NeurIPS).
>
> **Q3.* The overall idea of the article closely mirrors the KDD 2024 paper "Understanding Prompt Tuning for V-L Models Through the Lens of Neural Collapse" (Zhu et al., 2024). This similarity significantly diminishes the claimed originality of the paper, and there is insufficient comparison and explanation to clarify how the current work differs from the existing literature.
>
> * Answer:
>
> Thank you for your valuable feedback and for pointing out and highlighting the relevant work by Zhu et al., 2024. We appreciate the suggestion, and we will include this reference in our revised paper to provide a more comprehensive discussion of related works.
>
> While we acknowledge that the effectiveness of neural collapse in V-L models has been explored in prior works such as Zhu et al., 2024, their work primarily focuses on soft prompt tuning, which considers only whether text representations exhibit a simplex ETF-like structure. Their method relies on mitigating the discrepancy between image and text features via a mutual information loss $\mathcal{L}_{\text{MI}}$, assuming that image representations benefit from NC-like text features. Consequently, their framework provides limited control over the image features, and the class-wise constraint term and the class-wise constraint term $E_W$ is difficult to define and sensitive to the main loss function.
>
>
> In contrast, our work addresses a distinct problem: How to align the image feature with the NC structure while preserving the natural diversity of learned representations. We directly align the last-layer image features to the most suitable ETF prototypes (closest to the class description embeddings), and our proposed loss term $\mathcal{L}_{RSC}$ enables controllable representation pushing to enhance both data diversity and cross-class discrimination capability. This design enhances both intra-class compactness and inter-class separability, providing a more stable and interpretable alignment mechanism for few-shot image classification under the V-L paradigm.

---

> > ### Comment · Reviewer_HH2D · 2025-11-27
> >
> > The authors have indeed addressed some of my concerns; however, the high similarity to previous work still significantly weakens the paper's novelty. Therefore, I maintain my original score.

---

### Official Review · Reviewer_ijGe · 2025-10-29

**Soundness:** 2
**Presentation:** 3
**Contribution:** 2
**Rating:** 2
**Confidence:** 3

**Summary:**

This paper proposes SyNC, a CLIP-based few-shot training paradigm that balances fidelity and diversity when leveraging synthetic data. It combines a Neural Collapse loss that draws real and synthetic features toward shared ETF prototypes with a regional supervised contrastive loss that pushes apart misclassified synthetic examples. The method integrates with standard fine-tuning and reports gains over strong baselines—especially on fine-grained benchmarks—with ablations linking the fidelity–diversity balance to accuracy.

**Strengths:**

1. The paper tackles the real–synthetic training gap by explicitly balancing fidelity and diversity, coupling an ETF/Neural-Collapse–style alignment loss with a region-wise supervised contrastive loss.
2. The method is validated across multiple few-shot benchmarks and and ablations isolate each component.

**Weaknesses:**

1. The method feels like an engineering combo (ETF + Kabsch + region-wise contrast) rather than a new mechanism.
2. The average gain over ImagineFSL is small (~+0.4 pp) and mainly driven by FGVC-Aircraft, with several datasets underperforming; Table 1 also mixes configurations (Aircraft uses ETF-only while ETF+RSC is lower (Table 3)).
3. The paper lacks ablations on the role and quality of generated descriptions (vs. class names, length/structure, noise). Please add targeted ablations and qualitative examples plus simple quality metrics (e.g., CLIPScore/perplexity).
4. Many hyperparameters are introduced, yet the sensitivity plots don’t clearly link to the final gains; Kabsch brings little benefit except on FGVC-Aircraft.
5. There is no empirical complexity comparison for fine-tuning (latency, peak memory), leaving the cost–accuracy trade-off unclear.

**Questions:**

Please see the weaknesses.

---

> ### Author Response · Authors · 2025-11-21
> **Thanks for valuable suggestions**
>
> We thank the reviewer for your valuable suggestions and feedbacks. We would like to address you remaining concerns below.
> ## Weaknesses:
>
> **Q1.* The method feels like an engineering combo (ETF + Kabsch + region-wise contrast) rather than a new mechanism.
> * Answer:
>
> Thank you for your comment. We would like to highlight that our work addresses a fundamental problem observed in few-shot CLIP-based frameworks: how to balance the fidelity and diversity of feature representations when training with both real and synthetic data. We also emphasize the fundamental trade-off that previous methods only partially resolve, as outlined below.
>
> A critical challenge when training with synthetic data lies in balancing dataset quality and diversity. Specifically, the first challenge is matching the real and synthetic data distributions at both the pixel and feature representation levels. The second challenge is enhancing the diversity of synthetic data. Indeed, none of the existing methods adequately addresses both aspects of the problem, as stated in Lines 54–66.
>
> To tackle this challenge, we propose SyNC,that explicitly balances fidelity and diversity of feature representation when training with both real and synthetic data. For quality, we employ an ETF contrastive loss on both real and synthetic distributions to achieve two objectives: aligning the synthetic representations with the real data, and enhancing inter-class separation to improve classification performance. The Kabsch algorithm serves as a complementary component to ensure the quality of the ETF prototypes. For diversity, we propose region-wise contrastive learning as a controllable mechanism to prevent representational collapse toward the ETF prototypes, thereby maintaining broad coverage of the data representation space. To the best of our knowledge, SyNC represents a principled solution to a previously unaddressed problem in few-shot learning with synthetic data.
>
> **Q2.* The average gain over ImagineFSL is small (~+0.4 pp) and mainly driven by FGVC-Aircraft, with several datasets underperforming; Table 1 also mixes configurations (Aircraft uses ETF-only while ETF+RSC is lower (Table 3)).
>
> * Answer:
>
> We appreciate this detailed observation and would like to address several points. Firstly, we want to gently point out that ImagineFSL use a superior generator (Stable Diffusion 3) with all of the other remaining baselines including ours (Stable Diffusion 2.1), We included ImageFSL's results for reference, but the fairest comparisons are with DataDream and ProtoAug, which use identical settings to ours.
>
> We also want to clarify several important differences between SyNC and ImageFSL. ImageFSL employs substantially more resources than our method and other baselines: it uses Stable Diffusion 3 (superior version compared to SD v2.1 used in all other baselines), generates 300 images per class with  CLIPscore-based rejection sampling, and requires an additional nearly 500,000 images for self-supervised pre-training with a DINO-like architecture. While this rejection sampling helps filter synthetic images, it is highly sensitive to certain classes and often requires multiple generation attempts, increasing computational overhead. Moreover, despite ImageFSL's sophisticated prompt engineering pipeline using chain-of-thought with GPT-4 and LLaMA, controlling representation of synthetic data remains challenging, as evidenced by its weaker performance on fine-grained datasets (FGVC Aircraft and Stanford Cars) in Table 2.
>
> In contrast, SyNC offers a simpler and more efficient solution. We use basic "a photo of [class]" prompts in generating images (consistent with other baselines) while directly controlling representation through training. Morevoever, we fine-tune only the CLIP encoders, without the need of using additional DINO-like for  self-supervised training and prediction aggregation. Our method achieves comparable or slightly better performance across most datasets, providing a simpler solution to this problem.
>
> Regarding FGVC-Aircraft, this dataset is infamously known for its bad quality synthetic data by Stable Diffusion (v2.1) and by its fine-grained characteristic, making the decision boundary ambiguous as detailed in DataDream and ProtoAug. We believe that improve performance significantly on this dataset is also a meaningful contribution.
>
> Regarding the mixed configurations, we hypothesize that fine-grained datasets with small inter-class distances are highly sensitive to RSC loss, because it could push the representations of different classes become close to each other.  This is why we ablate different components and report dataset-specific optimal configurations. Across other datasets, SyNC consistently matches or exceeds ImageFSL (despite using an inferior generator) and substantially outperforms all methods using the same SD v2.1 generator.

---

> > ### Author Response · Authors · 2025-11-21
> >
> > **Q3.* The paper lacks ablations on the role and quality of generated descriptions (vs. class names, length/structure, noise). Please add targeted ablations and qualitative examples plus simple quality metrics (e.g., CLIPScore/perplexity).
> > * Answer:
> >
> > Thank you for this valuable suggestion. We conducted additional experiments to evaluate the quality of GPT-enhanced prompts versus simple "a photo of [class]" prompts using CLIP Score and Perplexity metrics. We used CLIP ViT-L/14 and open-source GPT-2 models for evaluation. The results are reported below:
> >
> > |Dataset|**CLIPScore**||**Perplexity**||
> > |-|-|-|-|-|
> > ||**"A photo of" Prompt**|**GPT Prompt**|**"A photo of" Prompt**|**GPT Prompt**|
> > |**EuroSAT**|0.2593|0.2459|2567.70|621.50|
> > |**Oxford Pets**|0.3128|0.3179|728.65|604.87|
> > |**DTD**|0.2681|0.2432|1294.37|158.81|
> > |**Caltech-101**|0.3103|0.2352|1193.11|428.71|
> > |**Flowers102**|0.2739|0.2695|1174.64|158.05|
> > |**Stanford Cars**|0.2828|0.2925|849.85|108.48|
> >
> > We observe that CLIP Scores are comparable between the two approaches, indicating similar image-text alignment quality. However, GPT-enhanced prompts show substantially lower perplexity across all datasets (4-16× improvement), validating their effectiveness in generating more natural and coherent descriptions. This improved linguistic quality helps correcting the initialization of Kabsch algorithm. We will include qualitative examples of generated descriptions and corresponding images in the revised manuscript.
> >
> >
> > **Q4.* Many hyperparameters are introduced, yet the sensitivity plots don’t clearly link to the final gains; Kabsch brings little benefit except on FGVC-Aircraft.
> > * Answer:
> >
> > We want gently point out that the hyperparameter λ, ρ, β, τ are fixed across all datasets. The only hyperparameters being tuned are the λ₁, λ₂ to control the ETF and RSC loss, and the learning rates and weight decays. Overall the number of tuned hyperparameters is equal or less than other current state-of-the-art methods like ProtoAUg and ImagineFSL. To clarify, we have provided the sensitivity analysis of λ₁, λ₂ in Table 6 and 7 in Appendix D. Regarding Kabsch alignment, while Table 4 shows modest average improvements, its contribution is very consistent across datasets.
> >
> > **Q5.* There is no empirical complexity comparison for fine-tuning (latency, peak memory), leaving the cost–accuracy trade-off unclear.
> > * Answer:
> >
> > We thank the reviewer for this suggestion. To demonstrate the cost-effectiveness of our method, we have conducted a detailed empirical analysis of training latency (clustering + fine-tuning) and peak memory usage on a single NVIDIA A100 GPU. We compare SyNC against the baseline (DataDream) and ProtoAug under identical settings (16-shot, 500 synthetic images per class).
> >
> > |Method|EuroSAT||||Pets||||Food101||||
> > |-|-|-|-|-|-|-|-|-|-|-|-|-|
> > ||Cluster|Train Time|Total|Acc|Cluster|Train Time|Total|Acc|Cluster|Train Time|Total|Acc|
> > |**DataDream**|-|13.3 min|13.3 min|93.4| -|50 min|50 min|94.5| -|133 min|133 min|87.5|
> > |**ProtoAug**|2.5 min|27.5 min|30 min|94.7|13 min|75 min|88 min|94.6|85 min|275 min|360 min|90.4|
> > |**SyNC$_{wETF}$**|-|23.5 min|23.5 min|94.5| -|92 min|92 min|95.2| -|230 min|230 min|90.4|
> > |**SyNC**|2.5 min|33 min|35.5 min|95.0|13 min|99 min|112 min|94.9|85 min|330 min|415 min|90.2|
> >
> > Note: The clustering time are independent from hyper-parameter fine-tuning step.
> >
> > While our full SyNC method incurs a computational overhead compared to DataDream (due to the K-Means clustering and geometry-aware loss calculation), it remains comparable to ProtoAug while achieving superior performance. Our simplified variant, SyNC$_{wETF}$, eliminates the clustering step entirely, offering a significant speedup over ProtoAug while still maintaining strong accuracy gains through the Neural Collapse loss.
> >
> > Regarding memory complexity, all methods were tested with the same backbone and batch size. We observed a consistent peak memory usage of ~10GB VRAM across all methods (DataDream, ProtoAug, and SyNC), ensuring the method remains easily reproducible on any standard hardware.

---

### Official Review · Reviewer_qjUR · 2025-10-30

**Soundness:** 3
**Presentation:** 2
**Contribution:** 2
**Rating:** 4
**Confidence:** 4

**Summary:**

This paper proposes SyNC, a new training paradigm for CLIP-based few-shot learning that aims to balance the fidelity and diversity of synthetic data representations. The method introduces two complementary losses:
(1) an ETF-based Neural Collapse loss to align real and synthetic features toward theoretically optimal geometric prototypes, thereby improving fidelity, and
(2) a regional supervised contrastive loss that enhances diversity by pushing apart misclassified synthetic features.
The framework integrates these components with LoRA fine-tuning on both real and generated samples. Experiments on ten benchmark datasets show competitive or superior performance, especially on fine-grained datasets such as FGVC-Aircraft and Stanford Cars.

**Strengths:**

- The paper is conceptually clear and easy to understand, providing a well-motivated formulation of the fidelity–diversity trade-off in synthetic-data-based few-shot learning.
- It accurately identifies the core limitation of prior works—overemphasis on either fidelity or diversity—and proposes a theoretically grounded direction to address both simultaneously.
- The integration of Neural Collapse theory into few-shot CLIP adaptation is novel and insightful, offering a principled geometric interpretation for feature alignment.

**Weaknesses:**

Although the paper is highly intuitive, the experimental evidence does not convincingly validate its hypotheses:
  1. The presentation of results in the tables can be **misleading**, as only the proposed method’s results are boldfaced even when other methods achieve identical result (see Tables 1, 3, and 4).
  2. In Table 1, SyNC’s results are not significantly better than ImagineFSL, which contradicts the claim in Lines 59–62 about "enhance diversity often pushes synthetic images further from the real data distribution"
  3. The ablation in Table 3 shows that the two proposed losses individually contribute marginal improvements on several datasets (e.g., IN, CAL, AirC, Pets, SUN, FLO). Furthermore, the paper’s justification in Section 3.1—“why not initialize ETF in a way better aligned with the input feature distribution?”—may be questionable, as the employed Kabsch algorithm merely aligns two point sets via least-squares rotation, functionally similar to random initialization followed by L2 fitting. This could explain the limited performance gains; thus, an error analysis would be necessary to substantiate the claimed benefits.
  4. The method involves too many hyperparameters (λ, λ₁, λ₂, ρ, β, τ), and the search ranges are inconsistent. For instance, the recommended range for λ₁ is [0, 1], yet in FGVC-Aircraft it is set to 15, far beyond the provided interval, suggesting a lack of systematic tuning.

**Questions:**

Please refer to Weaknesses.

---

> ### Author Response · Authors · 2025-11-21
> **Thank you for your supportive feedback**
>
> We would like to thank the reviewer for the valuablle suggestion and feedback. We address your remaining concerns below.
>
> **Q1.* The presentation of results in the tables can be misleading, as only the proposed method’s results are boldfaced even when other methods achieve identical result (see Tables 1, 3, and 4).
> * Answer:
>
> Thank you for raising this concern. We acknowledge that the current presentation of results in Tables 1, 3, and 4 may cause unintended ambiguity. This was not our intention, and we apologize for any confusion it may have caused.
>
> In the revised version of the paper, we will correct this by consistently boldfacing all methods that attain the same best performance. We will update the tables accordingly to ensure fairness and accurate comparison across methods.
>
> **Q2.* In Table 1, SyNC’s results are not significantly better than ImagineFSL, which contradicts the claim in Lines 59–62 about "enhance diversity often pushes synthetic images further from the real data distribution"
>
> * Answer:
>
> Thank you for raising this concern.  We would like to clarify several important differences between SyNC and ImageFSL. ImageFSL employs substantially more resources than our method and other baselines: it uses Stable Diffusion (SD) 3 (superior version compared to SD v2.1 used in all other baselines), generates 300 images per class with  CLIPscore-based rejection sampling, and requires an additional nearly 500,000 images for self-supervised pre-training with a DINO-like architecture. While this rejection sampling helps filter synthetic images, it is highly sensitive to certain classes and often requires multiple generation attempts, increasing computational overhead. Moreover, despite ImageFSL's sophisticated prompt engineering pipeline using chian-of-thought with GPT-4 and LLaMA, controlling representation of synthetic data remains challenging, as evidenced by its weaker performance on fine-grained datasets (FGVC Aircraft and Stanford Cars) in Table 2.
>
> In contrast, SyNC offers a simpler and more efficient solution. We use basic "a photo of [class]" prompts in generating images (consistent with other baselines) while directly controlling representation through training. Morevoever, we fine-tune only the CLIP encoders, without the need of using additional DINO-like for  self-supervised training and prediction aggregation. Our method achieves comparable or slightly better performance across most datasets, providing a simpler solution to this problem.
>
> **Q3.* The ablation in Table 3 shows that the two proposed losses individually contribute marginal improvements on several datasets (e.g., IN, CAL, AirC, Pets, SUN, FLO). Furthermore, the paper’s justification in Section 3.1—“why not initialize ETF in a way better aligned with the input feature distribution?”—may be questionable, as the employed Kabsch algorithm merely aligns two point sets via least-squares rotation, functionally similar to random initialization followed by L2 fitting. This could explain the limited performance gains; thus, an error analysis would be necessary to substantiate the claimed benefits.
>
> * Answer:
>
> Thank you for pointing out the need for more clarity regarding the difference between Kabsch algorithm's solution and the L2 fitting.
>
> Firstly, we would like to highlight that the Kabsch algorithm solution is exactly the closed-form solution of the L2 fitting. Consequenlty, we directly employ the intilization of the ETF structure as Kabsch algorithm.
>
> Secondly, we believe that using the Kabsch algorithm for ETF construction is supported by both theoretical justification and empirical evidence. As discussed in L206–L213, we emphasize that random initialization of the ETF structure (as seen in previous works [1,2,3]) can mitigate the model's performance. A key feature of our proposed SyNC method is that the Kabsch algorithm ensures prototypes always maintain a strong correlation with the embeddings of the classes they represent. This correlation ensures strong alignment between text and image features while preserving clear separation between classes. Additionally, the empirical results in Table 4 also demonstrate the consistent effectiveness of applying Kabsch rotation for ETF construction across 10 datasets.
>
> [1] Yang, Y., Chen, S., Li, X., Xie, L., Lin, Z., & Tao, D. (2022). Inducing Neural Collapse in Imbalanced Learning: Do We Really Need a Learnable Classifier at the End of Deep Neural Network? In Advances in Neural Information Processing Systems (NeurIPS 2022).
>
> [2] Yang, Y., Yuan, H., Li, X., Lin, Z., Torr, P., & Tao, D. (2023). Neural Collapse Inspired Feature‑Classifier Alignment for Few‑Shot Class‑Incremental Learning. In Proceedings of the 11th International Conference on Learning Representations (ICLR 2023).
>
> [3] Pham, T. D., Le, H., Ngo Van, L., Nguyen, N. T. N. D., Dinh, S., & Nguyen, T. H. (2025). Mitigating Non‑Representative Prototypes and Representation Bias in Few‑Shot Continual Relation Extraction. ACL.

---

> > ### Author Response · Authors · 2025-11-21
> >
> > **Q4.* The method involves too many hyperparameters (λ, λ₁, λ₂, ρ, β, τ), and the search ranges are inconsistent. For instance, the recommended range for λ₁ is [0, 1], yet in FGVC-Aircraft it is set to 15, far beyond the provided interval, suggesting a lack of systematic tuning.
> > * Answer:
> >
> > Thank you for your comment regarding the sensitivity analysis of hyperparameters.
> >
> > To clarify, the hyperparameter λ, ρ, β, τ are fixed across all datasets. We also want to note that the other baseline methods, such as ProtoAug and ImagineFSL, employ a comparable or more number of hyperparameters, suggesting that our design configuration is not unusually complex within this research context.
> >
> > Regarding the specific case of FGVC-Aircraft, this dataset is considered as the most challenging in this research field due to its highly fine-grained nature and the relatively poor quality of synthetic data produced by Stable Diffusion. Specially, we observed that increasing the weight λ₁ for the neural-collapse based loss substantially improved performance. The table below provides the results for the FGVC-Aircraft dataset under different parameter λ₁ settings, as evidenced by Table 6.
> >
> >
> > |λ₁|0.5|1|2|5|10|15|20|
> > |-|-|-|-|-|-|-|-|
> > |Accuracy|75.8|77.7|78.5|79.6|80.0|80.3|76.9|
> >
> > The results indicate that the fine-grained characteristics of FGVC-Aircraft make inter-class separation particularly crucial. Hence, strengthening the neural-collapse regularization helps enlarge inter-class margins, which directly contributes to improved performance on this dataset. Importantly, adjusting λ₁ in this way did not negatively affect performance on other datasets as can be see from the table 6.

---

### Meta-Review · Area_Chair_fAMZ · 2025-12-07

**Summary:**

The method addresses a good question with a clear presentation, but the novelty of NC for few-shot in CLIP is not convincing.
More experiments more than fine-grained scenarios are needed to validate its effectiveness over ImagineFSL.

**Reviewer Concerns:**

HH2D raised concern about the novelty on NC in CLIP.  YwNC raised concern on novelty of FL and NC combination.
qjUR and ijGe concerns on experimental validation is not convincing, such as results description and ablation.

**Reviewer Scores:**

All reviewers scores are below 4 with higher confidence.

---

### Decision · Program_Chairs · 2026-01-26

Reject